# Role of Lipids in Phosphine Resistant Stored-Grain Insect Pests *Tribolium castaneum* and *Rhyzopertha dominica*

**DOI:** 10.3390/insects13090798

**Published:** 2022-09-01

**Authors:** Ihab Alnajim, Naser Aldosary, Manjree Agarwal, Tao Liu, Xin Du, Yonglin Ren

**Affiliations:** 1Date Palm Research Center, University of Basrah, Basra 61004, Iraq; 2College of Science, Health, Engineering and Education, Murdoch University, Perth, WA 6150, Australia; 3Chemcentre, Resources and Chemistry Precinct, Bentley, WA 6102, Australia; 4Institute of Equipment Technology, Chinese Academy of Inspection and Quarantine, No. A3, Gaobeidianbeilu, Chaoyang District, Beijing 100123, China

**Keywords:** phosphine, insect resistance, *T. castaneum*, *R. dominica*, insect lipid, glycerolipids, phospholipids

## Abstract

**Simple Summary:**

The red flour beetle, *Tribolium castaneum*, and lesser grain borer, *Rhyzopertha dominica*, cause significant damage to stored commodities, including grains. The most widely used fumigant to control stored-product insects is phosphine. However, resistance to this fumigant is worldwide problem. In this study, we examined the lipid content of phosphine resistant and susceptible strains of *T. castaneum* and *R. dominica*. The results showed that the resistant strains of both species contained more lipids than the susceptible strains. The finding will contribute to the understanding of the mechanisms of phosphine resistance and provide additional information for developing strategies for managing the resistance problem.

**Abstract:**

Insects rely on lipids as an energy source to perform various activities, such as growth, flight, diapause, and metamorphosis. This study evaluated the role of lipids in phosphine resistance by stored-grain insects. Phosphine resistant and susceptible strains of the two main stored-grain insects, *Tribolium castaneum* and *Rhyzopertha dominica,* were analyzed using liquid chromatography-mass spectroscopy (LC-MS) to determine their lipid contents. Phosphine resistant strains of both species had a higher amount of lipids than susceptible stains. Significant variance ratios between the resistant and susceptible strains of *T. castaneum* were observed for glycerolipids (1.13- to 53.10-fold) and phospholipids (1.05- to 20.00-fold). Significant variance ratios between the resistant and susceptible strains of *R. dominica* for glycerolipids were 1.04- to 31.50-fold and for phospholipids were 1.04- to 10.10-fold. Glycerolipids are reservoirs to face the long-term energy shortage. Phospholipids act as a barrier to isolate the cells from the surrounding environment and allow each cell to perform its specific function. Thus, lipids offer a consistent energy source for the resistant insect to survive under the stress of phosphine fumigation and provide a suitable environment to protect the mitochondria from phosphine. Hence, it was proposed through this study that the lipid content of phosphine-resistant and phosphine-susceptible strains of *T. castaneum* and *R. dominica* could play an important role in the resistance of phosphine.

## 1. Introduction

The red flour beetle, *Tribolium castaneum* (Tenebrionidae: Coleoptera), and the lesser grain borer, *Rhyzopertha dominica* (Bostrichidae: Coleoptera), are stored-product insect pests that can cause serious damage to various commodities, including grain [1,2]. In addition to feeding on the grains, the damage by these species derives from contamination of the products with insect parts, ecdysis skin and individuals at each life stage [3], which can severely reduce grain quality and economic value [4].

Phosphine is one of the most widely used fumigants currently approved to control stored-product insect pests [5]. However, long-term use and ineffective applications have led to resistance to this fumigant by strains of stored-grain insect species, particularly *T. castaneum* and *R. dominica* [6,7]. Resistant insects usually absorb less phosphine than susceptible insects [8]. Suggesting that the exclusion of phosphine is a resistance mechanism [8]. Therefore, it was proposed that the metabolic and physiological variations between the resistant and susceptible strains are strongly associated with the resistance phenotypes [8,9,10]. A genetic study on dihydrolipoamide dehydrogenase (DLD) showed that it is a flavin-dependent oxidoreductase crucial for energy metabolism [11] and is important in phosphine resistance [9]. Dihydrolipoamide dehydrogenase comprises a reactive disulfide and a flavin adenine dinucleotide (FAD) cofactor directly involved in electron transfer in mitochondria [12]. As a by-product of its role in aerobic respiration, the DLD enzyme (rph2) generates reactive oxygen species (ROS), which contribute to phosphine toxicity to target insects [9]. In cellular membranes, fatty acid desaturase (FADS (rph1)) produces desaturated fatty acids that are targets of ROS [13]. Exposure to phosphine exacerbates ROS production, which damages fatty acids [9,13]. Therefore, rph1 and rph2 interact synergistically since FADS (rph1) sensitizes animals to ROS (13), whereas DLD (rph2) generates large amounts of ROS [14], which is exacerbated by phosphine exposure [5]. A homozygous for resistance alleles of rph1 reduces cellular membrane sensitivity to ROS, while homozygous for resistance alleles of rph2 reduces ROS production, resulting in extremely high levels of phosphine resistance [13].

Lipids comprise the largest component of some insect bodies, reaching 75% based on their dry weight [15]. The ability to store fat is essential for insects to adapt to their environment and undergo normal development and reproduction [16]. Additionally, lipids are structural components in cell membranes, have roles in intracellular signaling, and are the main reserved form of energy that insects use for diapause [17], growth [18], flight [19], and metamorphosis [16]. The composition of lipids is influenced by many factors, including genetic, ecological, and nutritional status, and varies across insect species [20]. Lipids are stored in the form of triglycerides (TGs) inside the fat bodies responsible for meeting the energy requirements of insects [21]. TGs can be stored in an anhydrous form, thereby allowing lipids as an essential substance for metabolism, enabling the accumulation of a large reservoir of energy that can be used during long periods of energy demand [22]. Diglycerides (DGs), on the contrary, are the major lipids in insect hemolymph [23]. The importance of DG was described in a study about locust flight when the amount of DG was increased in the hemolymph to threefold of its average concentration to supply the energy requirements [24,25]. Phospholipids are a large group of lipids that contain a polar and non-polar end, consisting of two layers: a hydrophobic layer that has two fatty acids and a hydrophilic layer of the phosphate group connected by glycerol or alcohol [26,27]. The significance of phospholipids is derived from their primary function as a significant part of cellular membranes, which acts as a barrier to isolate the cells from the surrounding environment and allows each cell to perform its specific function [28].

Current extensive research on pesticide resistance focuses on target site and metabolic resistance, but other mechanisms of resistance exist [29]. An example is modifying body parts in order to reduce insecticide penetration into the body, mainly by enhancing the deposit of structural components, such as epicuticular lipids [30]. As compared to pyrethroid-susceptible populations, pyrethroid-resistant *Triatoma infestans* had a significant increase (more than 50%) in cuticular lipid (CHCs), resulting in reducing the uptake of pyrethroids [31].

Both respiratory and metabolic factors may contribute to phosphine resistance [32]. Resistant strains have a lower respiratory rate than susceptible strains enabling resistant insects to receive less phosphine [33]. The importance of DLD in resistant strains is related to electron transfer resulting in enhancing the energy metabolism [9] and in regulating energy supply in resistant individuals [34]. The possibility of different metabolism between resistant and susceptible strains could lead us to explore the differences in the lipid composition between susceptible and resistant strains of *T. castaneum* and *R. dominica.* As mentioned above, lipids have a significant role in insecticide resistance by reducing the penetration of the insecticides into the target insect cells. Therefore, this study aimed to evaluate the role of lipids in two stored-grain insects to resist phosphine.

## 2. Materials and Methods

### 2.1. Insect Cultures

One-month-old adult insects were used in the experiments. Susceptible and resistant adult insects of *T. castaneum* (MUWTCSS-6000 and MUWTCSR) and *R. dominica* (MUWRDSS-7 and MUWRDSR-675) were obtained from the Department of Primary Industries and Regional Development (DPIRD), Perth, Australia, in 2016. The strains had been regularly treated with phosphine in the laboratory to promote homozygosity for phosphine resistance. Insects were cultured by incubating approximately 3000 2–3-day-old adults with 1000 g of food. The food for *T. castaneum* was wheat flour/yeast in a 12:1 ratio. The flour was made from newly harvested Australian standard wheat. To avoid contamination, the wheat was stored at −20 °C for 7 d before being transferred to storage at 4 °C until milling. The grain was milled with a Wonder Mill (Model WM2000, WonderMill Co., Seoul, South Korea), and the flour was stored at 4 °C until used. Before feeding to insects, the flour was brought up to room temperature overnight. For *R. dominica,* the food consisted of broken wheat. All cultures were maintained in 2 L glass jars sealed with mesh. Adult insects were allowed to mate and lay eggs for 4 days, after which they were removed, and the remaining cultural medium was incubated at 28 ± 1 °C, 70 ± 2% relative humidity (RH) and a photoperiod of 14:10 (L:D) h. As adult insects emerged, they were transferred to a new vessel to keep insects of similar ages together [10].

### 2.2. Preparation of Phosphine Gas for Determination of Resistance Factor

Phosphine was produced by adding Quickphos commercial tablets (United Phosphorus Limited Pty Ltd. (UPL), Adelaide, SA, Australia) of aluminum phosphide in 10% sulphuric acid solution to produce phosphine with final purity of 86% [35]. To obtain the resistance factor, bioassays were employed using the following concentrations of phosphine: 0, 0.005, 0.01, 0.02, 0.03, 0.04, 0.1, 0.3, 0.6, 1, 2, 3, and 4 mg/L, with three replicates of each concentration. Fifty adult insects in 1000 mL flasks for each replicate were fumigated with phosphine for 20 h for the susceptible and resistant strains, followed by a one-week recovery period at 25 ± 1 °C and 65 ± 5% RH. Based on probit model concentration-mortality analysis, the LC_50_ of 0.009 mg/L were considered the susceptible strain of both species, while the LC_50_ of 0.26 and 1.042 mg/L for *T. castaneum* and *R. dominica*, respectively, were considered as resistant strains [35]. Consequently, the resistance ratio was calculated according to LC_50_ of the susceptible insects (RR = 28.8-fold for *T. castaneum*) and (RR = 115.77-fold for *R. dominica*).

### 2.3. Chemicals and Apparatuses

The extraction and analysis of lipids were performed using the following materials: acetonitrile ≥99.9% *v*/*v* (Fisher Scientific, Glee, Belgium), methanol ≥99.9% *v*/*v*, 2-propanol ≥99.9% *v*/*v,* and chloroform ≥99.9% *v*/*v* (Sigma- Aldrich, Bellefonte, PA, USA). Apparatuses used were a 2 mL micro tube (Benchmark Scientific Inc., Sayreville, NJ, USA), 2 mL clear screw HPLC vials (Agilent Technologies, Santa Clara, CA, USA), different volumes of micropipettes (Dragon Laboratory Instruments Ltd., Beijing, China), bead bug micro tube homogenizer (Model DI030-E, Benchmark Scientific Ltd, Sayreville, NJ, USA), Dynamica Velocity 13µ micro centrifuge (Dynamica Pty Ltd., Mablethorpe, Lincolnshire, LN, United Kingdom), and an ultrasonic cleaner (Model PS-20A, Omegasonics, Simi Valley, CA, USA).

### 2.4. Extraction Procedures

All the insects used in this study were cleaned by allowing them to crawl on a wet tissue paper for 15 min, and then the insects were transferred into a clean dry tissue paper for 10 min. The cleaned insects were frozen to death and stored using liquid nitrogen. Adult insects were collected in a 2 mL micro tube using a small clean brush.

Fifteen adult insects of resistant and susceptible strains of *T. castaneum* and *R. dominica* were homogenized in 1 mL of chloroform/methanol (2:1, *v*/*v*) after adding three milling balls using the bead bug micro tube homogenizer at 400 rpm for 1 min [36]. The supernatant was filtered using a 3 mL syringe (Terumo Australia Pty Limited (TAUS), Sydney, NSW Australia) coupled with 13 mm 0.2 µm Agilent Captiva Econo Filters (Agilent Technologies, Santa Clara, CA, USA). To separate the non-lipids substances, the filtered homogenate was washed with 0.2 mL distilled water and centrifuged at 2000 rpm using Dynamica Velocity micro centrifuge. After centrifuging, the upper phase, which contains non-lipids substances, was removed. The lower chloroform phase, which contains lipids, was transferred into a 2 mL GC clear vial, which was already weighed as (W_vial_). The extract was blown to dryness under nitrogen flow. The same vial was weighed as (W_vial+ lipids_) for calculating total lipids weight (W_lipids_) according to the following equation (Equation (1)):W_lipids_ = W_vial+ lipids_ − W_vial_(1)

After calculating the total lipids, 600 µL of HPLC solvent (Isopropanol/Acetonitrile/Water (2:1:1, *w*/*w*/*w*) was added to reconstitute the dried lipid components for UPLC-Q-ToF analysis. The ultrasonic cleaner was also used to assist the dissolution of the lipids.

### 2.5. Alalysis of Lipids with Ultra Performance Liquid Chromatography-Quadrupole-Mass Spectrometry (UPLC-Q-ToF-MS) and Analytical Conditions

Samples were analyzed using Waters Acquity UPLC-Q-Tof. Data acquisition and processing were performed using the Masslynx software (version 4.1, Waters Corporation, Milford, MA, USA).

For analysis of lipids, separation of lipid compounds was performed on a Waters Acquity BEH C18 column (2.1 × 100 mm, 1.7 μm). The binary gradient consisted of eluents A (60% acetonitrile: 40% water *w*/*w*) and B (90% 2-propanol: 10% acetonitrile *w*/*w*) with 10 µM ammonium formate and 0.1% formic acid at room temperature with a flow rate at 0.25 mL/min and a 2 μL injection volume. Optimal separation was accomplished using the following solvent gradient elution: mobile B started with 40%, increased to 92.1% (1−16 min), then ramped back to 40% (17–17.5 min), followed by 2.5 min of re-equilibration with a total run time of 20 min. All features were analyzed in a positive ionization mode and were monitored in a full scan mode. The optimum MS parameters were capillary voltage 3.1 kV, sample cone 45 V, extraction 5.0 V, ion guide voltage 3.0 V, desolvation gas temperature 350 °C with 350 L/min, collision cell 0.6 mL/min of UHP Argon, and detector voltage 1820 V.

### 2.6. Data Processing and Analysis

All measurements were conducted according to completely randomized design (CRD). All the samples were analyzed in four biological replicates. The LC-MS data samples were analyzed as one batch to ensure that the parameters would be applied equally in all the samples. Peak deconvolution, filtering, scaling, and integration were extracted and aligned using the MassLynx software (version 4.1, Waters Corporation, Milford, MA, USA). Chromatographic peaks were extracted from 1 to 20 min with a retention time error window of 0.2 min, and the mass spectral peaks detected ranged from 50 to 2000 *m*/*z* with a mass error window of 7 ppm. The resulting data matrix extracted from total ion consisted of retention time, and *m*/*z* was generated together with peak intensity based on peak area for all features.

The mass spectra of the lipids were loaded into LIPID MAPS Lipidomics Gateway (http://www.lipidmaps.org/tools/ms/lmmassform.php, accessed on 30 August 2018). The identification search was restricted to two main lipids categories, which included glycerolipids and phospholipids. The following parameters were applied for an appropriate identification: mass tolerance ±0.1 *m*/*z* and ion adducts of positive mode [M+H]+ and [M+NH4]+. The loaded spectra were compared with the matched spectra, which were obtained from the lipid maps to identify the compounds. The compounds with the highest spectrum match factor were chosen as the lipid compound candidates.

Data were normalized with internal standards before statistical evaluation. Statistical analysis was employed to evaluate and visualize the data through MetaboAnalyst 4.0 (https://www.metaboanalyst.ca/MetaboAnalyst/upload/StatUploadView.xhtml, accessed on 30 August 2018) using volcano plot analysis and *t*-test [37]. Samples were uploaded to Metaboanalyst as columns (unpaired); data filtering was characterized by using the mean intensity value. Sample normalization, data transformation, and data scaling were specified as a “NONE” mode. Volcano plot was analyzed at a *p*-value threshold of 0.05 and fold change threshold ≥2. Figure 1 was generated using IPM SPSS statistics 24 (Murdoch University version).

## 3. Results and Discussion

The total lipid contents of susceptible and resistant strains of *T. castaneum* and *R. dominica* were evaluated according to the Floch method [36]. The results showed a significant difference in the quantity of total lipids (Figure 1). The resistant strains of both *T. castaneum* and *R. dominica* species contained a significantly greater amount (*p* values <0.001 for *T. castaneum* and <0.01 for *R. dominica*) of lipids compared to the susceptible strains of both species (Figure 1).

Lipid samples from both susceptible and resistant strains of insects were tested to determine the differences in phospholipids and glycerolipids for their predicted role in phosphine resistance. Variances were observed in relation to the major peaks of LC-MS base peak intensities chromatograms when comparing susceptible and resistant strains of both insect species. A comparison of the lipids data obtained from the profiles of susceptible and resistant insects of the two studied species is shown in Figure 2 and Figure 3. The main difference between the susceptible and resistant strains of *T. castaneum* in the chromatograms was between RT = 6.67 to 10.64 min (Figure 2). In contrast, the differences between the two strains of *R. dominica* included the majority of the peaks in the base peak intensity chromatogram (Figure 3).

The lipids separated from the insect samples were further characterized by MS detector. In the identification of the lipid compounds, only glycerolipids and phospholipids were studied for their role in the energy production and structuring of the cell membranes. The fold changes results (using volcano plot statistical analysis) revealed a higher quantity of lipids obtained from resistant strains than susceptible insects in both species. High difference ratios were obtained for most lipids obtained from *T. castaneum,* ranging for glycerolipids from 1.13- to 53.10-fold and phospholipids from 1.05- to 20.00-fold (Table 1). In comparison, the fold changes in glycerolipids for *R. dominica* were between 1.04- to 31.50-fold and for phospholipids from 1.04- to 10.10-fold (Table 2).

A total of 45 features from *T. castaneum* and 67 from *R. dominica* were identified as lipids belonging to either glycerolipids or phospholipids (Table 1 and Table 2). The statistical analysis revealed significant differences between the two lipids categories. The resistant insects contained more lipid compounds in abundance in both lipid categories than the susceptible strains (Table 1 and Table 2).

Volcano plot statistical analysis identified 17 glycerolipid and 8 phospholipid features from *T. castaneum* as significant (*p*-value ≤ 0.05 and fold change ≥ 2). The highest significant difference between resistant and susceptible strains for the glycerolipids was recorded for the compound’s ID 6.89_782.485 (*p*-value < 0.0005). In contrast, the compound’s ID 7.77_758.494 (*p*-value < 0.0005) recorded the highest significant difference for the phospholipids. A statistical analysis of volcano plot selected 18 features associated with glycerolipids and 8 associated with phospholipids as significant features in comparing resistant and susceptible strains of *R dominica*. The compound’s ID 14.48_874.714 recorded the highest significant difference between resistant and susceptible strains in the glycerolipids category (*p*-value < 0.0005). The compound’s ID 7.81_806.504 recorded the highest significant difference in the phospholipid category (*p*-value < 0.0005). In addition, some other lipids were significantly higher in resistant individuals than the susceptible strains in both *T.*
*castaneum* and *R. dominica.* These included lipid IDs 6.89_782.485 (DGDG(23:1)), 8.77_786.518 (MGDG(36:9)), 8.48_760.505 (PG(34:4)), 8.73_577.469 (DG(33:3)), 7.27_599.455 (DG(35:6)), 7.20_740.45 (MGDG(32:4)), 8.03_575.455 (DG(33:4)), 6.88_601.468 (MGDG(23:2)), 8.77_718.469 (PC(30:2(OH))), and 14.06_900.704 (TG(55:11)) (Table 1 and Table 2). 

Our examination of total lipid content aimed to provide an overview of the variances of the lipid amounts presented in insect bodies of phosphine-resistant and phosphine-susceptible strains of two insect species. The greater amount of lipids in phosphine-resistant insects of both *T. castaneum* and *R. dominica* led us to hypothesize that lipids may have a significant survival role with regard to phosphine resistance. We considered the importance of lipids for resistant insects as a factor required to tolerate more effectively the toxic effect of phosphine. We also considered the strong link that was reported between greater amounts of lipids, specifically cuticular lipids, and higher resistance to pesticides in a variety of insect species, such as *Drosophila melanogaster* to DDT [38] and *Triatoma infestans* to pyrethroids [31].

Insects rely more on lipids in severe long-term conditions, such as exposure to lack of energy [39]. This is because lipids are reserved for recovering from the lack of energy for long periods [20]. Strains with high-energy demands might be more susceptible to phosphine because of the increased mitochondrial activity levels that are necessary to sustain energy production [9]. This observation is because phosphine, a respiratory inhibitor in the mitochondria of insects and rats [40,41], disturbs the energy production of mitochondria [42]. The inhibitory effect on the mitochondria explains why lack of energy is one of the plausible reasons for mortality in insects due to phosphine [32,42]. This result is consistent with research results that showed that artificially raising the energy demand increased the sensitivity of the nematode *Caenorhabditis elegance* toward phosphine [43]. Thus, more lipids in resistant insects can provide survival factors to resist the negative effect of phosphine, that is, a decrease in energy by affecting mitochondria.

In our study, the outcome further supports the hypothesis that increased lipids provide advantages to survive the toxic effect of phosphine. The negligible impact of phosphine on resistant insects compared to susceptible insects indicated that metabolic factors contribute to resistance to phosphine [44]. A study on roundworm *C. elegans* revealed that phosphine affected both structure and function of mitochondria; however, phosphine-resistant *C. elegans* had a substantial increase in the mitochondrial membrane potential and less oxygen consumption (43). Consistent with that, reduced levels of respiration were acquired for resistant strains with high resistance ratios from different insect species, such as *T. castaneum, R. dominica*, and *O. surinamensis* (33). It may be that the higher mitochondrial membrane potential gives resistant strains an advantage in using energy resources to avoid the effect of phosphine.

The results indicate an increase in glycerolipids, which are considered major energy sources. Glycerolipids contain triglycerides, which, along with glycogen, are the main sources of stored energy in insect bodies [45]. Glycogen is consumed in the short-term [46]. While triglycerides have a higher caloric content than glycogen and are the main source of releasing fatty acids, which can be used for energy production [47]. Stored fatty acids are utilized in different forms for many purposes, such as energy provision to perform metabolic activities [20]. Another advantage of lipids is that fatty acids play a role in synthesizing energy components, such as trehalose [47] and proline, which is oxidized during endothermic pre-flight warm-up and during flight after prolonged starvation [48]. That is why higher concentrations of triglycerides that were found in this study may help resistant insects avoid the effect of phosphine on the mitochondria, which causes a reduction in the energy that causes death.

Significantly higher content of diglycerides obtained from different metabolic pathways in resistant insect strains were compared with susceptible strains, as this plays an essential role in being the main source for triglycerides synthesis [20]. Diglycerides are also important because they are the core lipids in insect hemolymph after the triglycerides degradation [49]. According to their importance, as explained above, the significant differences in the triglycerides and diglycerides between the resistant and susceptible strains obtained in this study indicate that these compounds are being utilized by the insects to survive from phosphine exposure, especially after long-term exposure.

Phospholipids levels were significantly higher in the resistant insects than in the susceptible insects. This finding supports the assumption that resistant insects rely on lipids to survive the phosphine effect. Phospholipids exist in organisms as essential compounds to maintain life activity and are also vital components of cellular and semi-cellular membranes [27]. Furthermore, they are crucial parts of cellular membranes, which act as barriers between cells and their surrounding environment and enable each cell to perform its specific function [28]. This characteristic may reduce or prevent phosphine penetration to the cells, thereby causing more exclusion of phosphine. This is consistent with one of the accepted explanations that the exclusion of phosphine is a possible resistance mechanism [8].

Additionally, phospholipids are essential for improving nerve cell function [50]. Moreover, phosphine also affects the neural system [51]; therefore, the higher concentration of phospholipids in resistant insects may improve the functions of the neural system in these insects. Furthermore, phospholipid of the mitochondrial membrane that is rich in unsaturated fatty acids plays an essential part in mitochondrial energy by affecting the activity of proteins of the mitochondrial inner membrane [52]. Phospholipids also contribute a significant amount to the mitochondrial membrane lipid environment. They have a substantial role in the mitochondrial respiratory chain by affecting the physical properties of the mitochondrial membrane [53]. Hence, respiration was found to be affected by the reduction in the mitochondrial phospholipids [54]. Therefore, a higher concentration of phospholipids in resistant strains may enhance mitochondria function and reduce the impact of phosphine toxicity.

In addition to its effect on the mitochondrial respiratory chain, the reduction in phospholipids was also observed to be synchronous with a significant reduction in adenosine triphosphate (ATP) [44]. A study by Price and Walter [55] on lesser grain borer *R. dominica,* demonstrated that ATPs were reduced from 2.75 to 1.64 nmoles/insect after treating the insect with phosphine. Phosphine causes a severe reduction in cytochrome oxidase activities and affects nicotinamide adenine dinucleotide (NAD) and succinic dehydrogenase activities, thereby leading to a reduction in respiration and causing a decline in the synthesis and ATP level [42]. Therefore, improving the mitochondrial energy and raising the mitochondria function by a higher content of phospholipids will certainly affect phosphine toxicity and energy production. Another advantage of the higher levels of phospholipids is increasing the production of phosphatidic acids from the glycerophosphate pathway. Phosphatidic acids are also considered as one of the main sources of triglycerides composition that allows the presence of more energy sources [20], which can provide more energy to the resistant insects to resist phosphine.

Finally, as reported in previous studies, both rph1 and rph2 contribute to phosphine toxicity by damaging fatty acids [9,13]. In resistant insects, which have homozygous for resistance alleles of rph1 or rph2, the damage of fatty acids is extremely reduced due to the reduction in the sensitivity of cell membranes to reactive oxygen species (ROS) [13]. This, in turn, might lead to an abundance of fatty acids, which are the primary components of lipid formation.

## 4. Conclusions

The levels of lipids and contents were investigated and compared between phosphine-resistant and -susceptible strains of *R. dominica* and *T. castaneum*. The total lipid content was found to be higher in the resistant strains than in the susceptible strains. Results showed that most glycerolipids and phospholipids in the resistant insects were more abundant than in the susceptible insects. Both glycerolipids and phospholipids play a significant role in tolerating the harmful effect of phosphine in phosphine-resistant insects by their contribution to providing energy sources and building cell walls. This research will benefit in developing a strategy for managing phosphine resistant insect pests based on understanding the role of lipids in phosphine resistance of the stored-grain insect pests, *T. castaneum* and *R. dominica*. We recommend conducting further studies on isogenic phosphine-resistant and -susceptible strains of insects.

## Figures and Tables

**Figure 1 insects-13-00798-f001:**
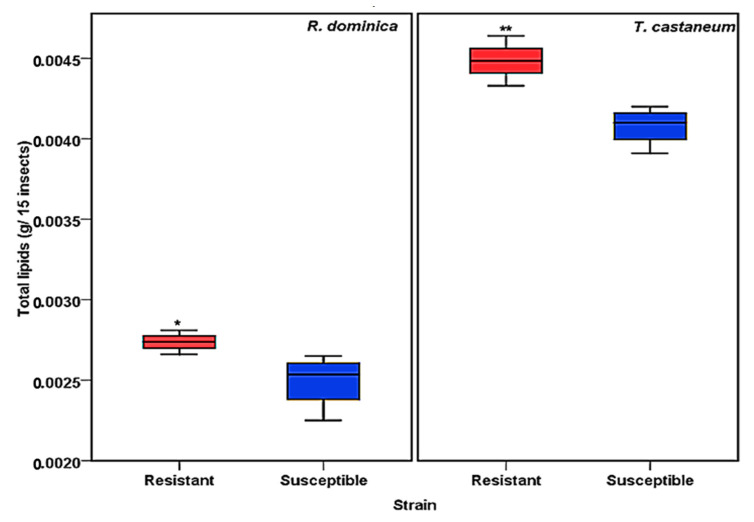
Total lipid content obtained from phosphine-susceptible and -resistant strains of *R. dominica* and *T. castaneum*. Each value in the figure represents the average of four biological insect sets. The values were statistically analyzed by *t*-test. * = *p* ≤ 0.01, ** = *p* ≤ 0.001.

**Figure 2 insects-13-00798-f002:**
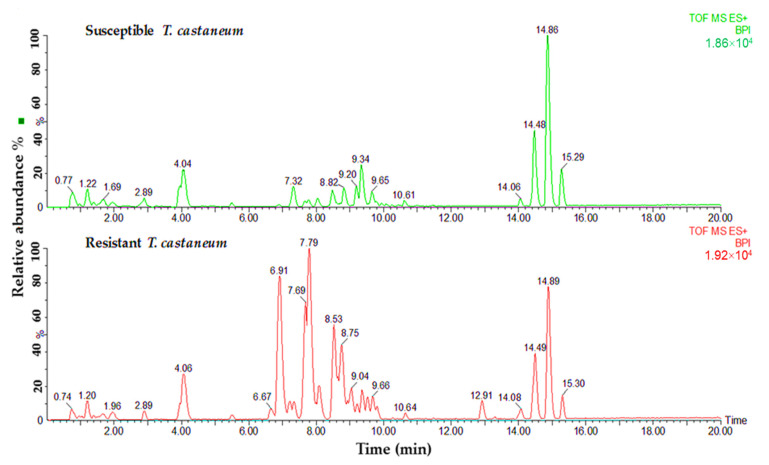
Base peak intensity (BPI) chromatograms show the differences in the lipid content obtained from susceptible (green) and resistant (red) strains of *T. castaneum*.

**Figure 3 insects-13-00798-f003:**
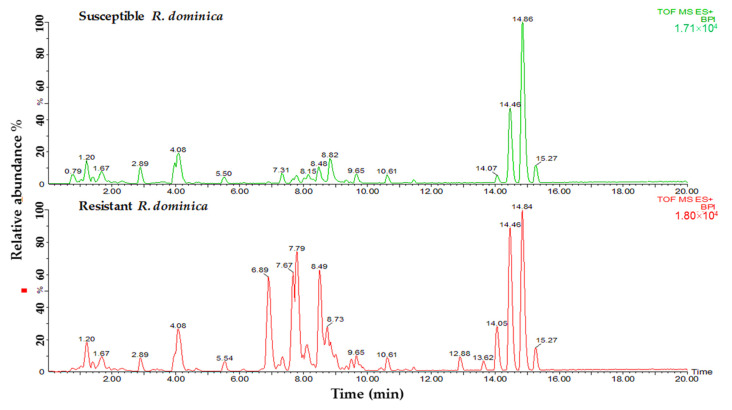
Base peak intensity (BPI) chromatograms show the differences in the lipid content obtained from susceptible (green) and resistant (red) strains of *R. dominica*.

**Table 1 insects-13-00798-t001:** Glycerolipids and phospholipids candidates obtained from *T. castaneum*.

No	Lipids ID	Input Mass	Matched Mass	Lipid Category	Name	Ion	LC-MS Response (n = 4)		FC	*p* Value
							Resistant	Susceptible		
1	1.99_369.3	369.3532	369.3727	phospholipids	LPA(13:0)	[M+H]+	5.02 ± 0.83	2.31 ± 1.44	2.17	<0.05
2	1.99_522.3	522.2812	522.2826	phospholipids	LPC(18:1)	[M+NH4]+	99.91 ± 16.34	49.16 ± 7.18	2.03	<0.0005
3	1.99_544.2	544.3822	544.3609	phospholipids	LPC(20:4)	[M+NH4]+	31.13 ± 4.84	43.99 ± 6.80	0.70	ns
4	4.04_621.2	621.3114	621.3034	phospholipids	LPI(20:4)	[M+H]+	832.66 ± 52.05	790.97 ± 101.41	1.05	ns
5	4.04_666.2	666.3628	666.3613	phospholipids	LPI(22:4)	[M+NH4]+	68.50 ± 8.76	61.30 ± 13.01	1.11	ns
6	5.60_457.2	457.2211	457.2561	phospholipids	LPG(14:0)	[M+H]+	2.00 ± 0.56	1.09 ± 0.13	1.84	ns
7	5.60_437.3	437.3476	437.3625	glycerolipids	MG(24:3)	[M+H]+	7.89 ± 0.73	4.95 ± 0.90	1.59	ns
8	6.88_782.4	782.4642	782.4896	glycerolipids	DGDG(23:1)	[M+NH4]+	2717.16 ± 44.73	54.27 ± 5.57	50.10	<0.0005
9	6.88_804.4	804.4523	804.474	glycerolipids	DGDG(25:4)	[M+NH4]+	133.56 ± 4.30	4.31 ± 1.27	31.00	<0.0005
10	6.88_701.4	701.4454	701.4259, 701.5715	glycerolipids	MGDG(31:8), TG(41:4)	[M+H]+	74.04 ± 4.77	45.81 ± 9.18	1.62	ns
11	6.88_601.4	601.4694	601.4826, 601.4979	glycerolipids	MGDG(23:2), DG(35:5)	[M+H]+	125.56 ± 9.54	11.98 ± 1.26	10.50	<0.0005
12	7.20_740.4	740.5026	740.5307	glycerolipids	MGDG(32:4)	[M+NH4]+	338.85 ± 42.19	6.39 ± 0.85	53.10	<0.0005
13	7.27_599.4	599.4126	599.379, 599.467	glycerolipids	MGDG(23:3), DG(35:6)	[M+H]+	159.43 ± 10.07	3.36 ± 0.63	47.50	<0.0005
14	7.22_703.4	703.4723	703.4416, 703.5871	glycerolipids	MGDG(31:7), TG(41:3)	[M+H]+	273.52 ± 50.17	354.10 ± 23.76	0.77	ns
15	7.75_784.4	784.4761	784.5053	glycerolipids	DGDG(23:0)	[M+NH4]+	3137.94 ± 229.11	117.26 ± 8.82	26.80	<0.0005
16	7.75_758.4	758.4732	758.4967, 758.4814	phospholipids	PG(34:5), PI(27:0)	[M+NH4]+	2087.66 ± 76.10	104.59 ± 13.47	20.00	<0.0005
17	7.77_806.4	806.4587	806.4814	phospholipids	PS(38:7), PC(37:7(OH))	[M+NH4]+	110.65 ± 13.14	8.09 ± 0.64	13.70	<0.0005
18	7.77_603.4	603.4434	603.4103, 603.4983	glycerolipids	MGDG(23:1), DG(35:4)	[M+H]+	313.60 ± 17.51	36.05 ± 4.00	8.70	<0.0005
19	8.04_729.4	729.4751	729.4701, 729.6028	phospholipids	PE(38:8), PS(34:2)	[M+H]+	306.08 ± 6.74	176.42 ± 29.97	1.74	ns
20	8.03_575.4	575.453	575.467	glycerolipids	DG(33:4)	[M+H]+	158.93 ± 7.77	18.01 ± 3.30	8.82	<0.0005
21	8.48_760.5	760.4868	760.5123,	phospholipids	PG(34:4)	[M+NH4]+	1561.83 ± 22.12	172.36 ± 20.44	9.06	<0.0005
22	8.52_731.4	731.4872	731.5668, 731.6184	glycerolipids	MGDG(32:0), TG(43:3)	[M+H]+	362.89 ± 10.73	319.09 ± 50.36	1.13	ns
23	8.77_786.5	786.4706	786.427, 786.5151	glycerolipids	DGDG(24:6),MGDG(36:9)	[M+NH4]+	1827.19 ± 44.90	106.02 ± 17.82	17.14	<0.0005
24	8.83_663.3	663.3986	663.4103, 663.4983	glycerolipids	MGDG(28:6), DG(40:9)	[M+H]+	359.03 ± 23.18	296.30 ± 23.71	1.21	ns
25	8.77_718.4	718.4232	718.4525, 718.5017	phospholipids	PC(30:2(OH)),PE(33:2(OH))	[M+NH4]+	287.33 ± 35.33	29.15 ± 6.48	9.86	<0.0005
26	8.83_664.3	664.3537	664.3456	phospholipids	LPI(22:5), PS(27:1)	[M+NH4]+	154.02 ± 1.65	132.65 ± 10.12	1.16	ns
27	8.73_577.4	577.4341	577.3946, 577.4826	glycerolipids	MGDG(21:0), DG(33:3)	[M+H]+	174.22 ± 10.17	21.30 ± 2.10	8.18	<0.0005
28	9.03_744.4	757.4978	757.5824, 757.6341	glycerolipids	MGDG(34:1), TG(45:4)	[M+H]+	704.50 ± 65.61	29.31 ± 6.03	24.00	<0.0005
29	9.04_604.4	604.4458	604.4055, 604.4935	glycerolipids	MGDG(22:2), DG(34:5)	[M+NH4]+	147.95 ± 7.35	6.72 ± 1.56	22.00	<0.0005
30	9.37_772.4	772.4944	772.5933, 772.64	glycerolipids	MGDG(34:2), TG(45:5)	[M+NH4]+	470.23 ± 76.31	749.62 ± 135.62	0.63	ns
31	9.37_795.4	795.4863	795.5042, 795.51	glycerolipids	MGDG(38:10),DGDG(25:0)	[M+H]+	14.07 ± 2.71	24.61 ± 2.25	0.57	ns
32	9.37_762.4	762.4745	762.5151, 762.427	glycerolipids	MGDG(34:7), DGDG(22:4)	[M+NH4]+	25.54 ± 1.07	12.98 ± 0.96	1.97	ns
33	9.80_931.5	931.5	931.5331	phospholipids	PI(42:10)	[M+H]+	103.54 ± 11.13	8.10 ± 0.99	12.8	<0.0005
34	9.80_752.5	752.50	752.5072, 752.5225	phospholipids	PE(36:6(OH)), PC(34:5)	[M+H]+	201.86 ± 18.26	17.15 ± 4.22	11.80	<0.005
35	12.93_892.6	892.6547	892.7389	glycerolipids	TG(54:8)	[M+NH4]+	48.15 ± 9.55	9.74 ± 1.50	4.95	<0.0005
36	13.30_895.6	895.6443	895.681	glycerolipids	TG(56:12)	[M+H]+	41.92 ± 7.31	3.86 ± 3.73	10.90	<0.005
37	14.06_900.6	900.6378	900.7076	glycerolipids	TG(55:11)	[M+NH4]+	101.13 ± 2.30	15.72 ± 2.58	6.43	<0.05
38	14.49_874.6	874.6416	874.6919	glycerolipids	TG(53:10)	[M+NH4]+	1566.44 ± 180.26	868.03 ± 114.93	1.80	ns
39	14.49_848.6	848.6331	848.6763	glycerolipids	TG(51:9)	[M+NH4]+	909.28 ± 89.13	347.89 ± 51.29	2.61	<0.05
40	14.49_822.6	822.6191	822.6606	glycerolipids	TG(49:8)	[M+NH4]+	293.52 ± 16.92	118.59 ± 17.31	2.48	<0.05
41	14.90_876.6	876.6555	876.7076	glycerolipids	TG(53:9)	[M+NH4]+	3310.62 ± 139.19	2189.69 ± 296.72	1.51	ns
42	14.90_850.6	850.644	850.6919	glycerolipids	TG(51:8)	[M+NH4]+	2272.31 ± 153.12	1155.83 ± 164.14	1.97	ns
43	14.90_902.6	902.6696	902.7232	glycerolipids	TG(55:10)	[M+NH4]+	474.79 ± 64.08	212.74 ± 22.57	2.23	<0.05
44	15.32_878.6	878.6759	878.7232	glycerolipids	TG(53:8)	[M+NH4]+	838.12 ± 60.35	391.75 ± 57.23	2.14	<0.05
45	15.32_904.6	904.6899	904.7389	glycerolipids	TG(55:9)	[M+NH4]+	370.22 ± 57.50	145.12 ± 24.04	2.55	<0.05

Lipids ID = contains retention time and mass spectrum of the lipid, Input Mass = mass spectra obtained from LC-MS, Matched mass = mass spectra obtained from LIPID MAPS database, Ion = LC-MS run mode, MG = monoglycerol, DG = diglycerol, TG = triglycerol, MGDG = monogalactosyldiacylgylcerol, DGDG = digalactosyldiacylgylcerol, PG = phosphtatidylglycerol, PI = phosphtatidylinositol, PS = phosphtatidylserine, PC = phosphtatidylcholine, PE = phosphtatidylethanolamine, LPG = lysyl-phosphatidylglycerol, LPI = lysophosphatidylinositol, LPA = lysophosphatidic acid. Values with FC (fold change) less than two were considered not significant (ns) by volcano plot statistical analysis.

**Table 2 insects-13-00798-t002:** Glycerolipids and phospholipids candidates obtained from *R. dominica*.

No	Lipids ID	Input Mass	Matched Mass	Lipid Category	Name	Ion	LC-MS response (n = 4)		FC	*p* Value
							Resistant	Susceptible		
1	1.20_333.2	331.2029	331.2843	glycerolipids	MG(16:0)	[M+H]+	494.20 ± 39.19	373.22 ± 30.01	1.32	ns
2	1.42_372.2	372.2362	372.3108	glycerolipids	MG(18:2)	[M+NH4]+	165.21 ± 8.96	123.47 ± 11.27	1.34	ns
3	1.42_504.3	504.3135	504.4047	glycerolipids	MG(28:6)	[M+NH4]+	194.33 ± 5.74	150.90 ± 5.74	1.29	ns
4	1.42_522.3	522.3278	522.4153	glycerolipids	DG(28:4)	[M+NH4]+	133.57 ± 9.96	79.67 ± 15.17	1.68	ns
5	1.77_497.3	497.3189	497.2874	phospholipids	LPG(17:1)	[M+H]+	55.96 ± 3.28	79.22 ± 5.13	0.71	ns
6	2.85_440.3	440.3844	440.4462	phospholipids	LPA (P-18:0)	[M+NH4]+	131.61 ± 38.07	228.69 ± 74.20	0.58	ns
7	2.85_387.2	387.2717	387.3469	glycerolipids	MG(20:0)	[M+H]+	22.74 ± 2.45	26.03 ± 0.93	0.87	ns
8	2.85_397.3	397.3575	397.404	phospholipids	LPA (O-16:0)	[M+H]+	8.03 ± 2.17	7.60 ± 2.14	1.06	ns
9	2.85_560.3	560.3673	560.283	phospholipids	LPI(14:1)	[M+NH4]+	22.75 ± 1.48	26.00 ± 0.18	0.88	ns
10	4.006_621.2	621.2667	621.3034	phospholipids	LPI(20:4)	[M+H]+	875.73 ± 15.65	764.82 ± 134.20	1.15	ns
11	4.006_468.4	468.4175	468.4775	phospholipids	LPA (P-20:0)	[M+NH4]+	147.30 ± 42.49	142.10 ± 46.77	1.04	ns
12	6.91_782.5	782.504	782.4896	glycerolipids	DGDG(23:1),	[M+NH4]+	2019.82 ± 235.61	68.25 ± 17.86	29.60	<0.0005
13	6.91_701.5	701.5061	701.5715,701.4259	glycerolipids	TG(41:4), MGDG(31:8)	[M+H]+	48.53 ± 7.57	27.33 ± 4.27	1.78	ns
14	6.91_805.4	805.4935	805.4944	glycerolipids	DGDG(26:2)	[M+H]+	48.53 ± 7.57	27.33 ± 4.27	1.72	ns
15	6.91_601.4	601.4752	601.4826,601.3946	glycerolipids	DG(35:5), MGDG(23:2)	[M+H]+	122.02 ± 15.25	10.84 ± 3.47	11.30	<0.0005
16	7.32_703.5	703.5203	703.5355,703.5871	glycerolipids	MGDG(30:0), TG(41:3)	[M+H]+	274.50 ± 37.66	197.51 ± 29.57	1.39	ns
17	7.32_740.4	740.4688	740.5307	glycerolipids	MGDG(32:4)	[M+NH4]+	200.90 ± 26.49	14.97 ± 3.73	13.40	<0.0005
18	7.32_599.4	599.4777	599.4823, 599.379	glycerolipids	MGDG(23:3), DG(35:6)	[M+H]+	98.52 ± 11.18	6.03 ± 0.87	16.30	<0.0005
19	7.32_725.4	725.4998	725.5198	glycerolipids	MGDG(32:3)	[M+H]+	21.10 ± 2.32	16.90 ± 2.12	1.25	ns
20	7.73_603.4	603.495	603.4983,603.4103	glycerolipids	DG(35:4), MGDG(23:1)	[M+H]+	345.30 ± 25.45	29.48 ± 8.50	11.70	<0.0005
21	7.73_806.5	806.5099	806.4814	phospholipids	PS(38:7), PC(37:7(OH))	[M+NH4]+	100.43 ± 4.92	11.88 ± 1.87	8.45	<0.0005
22	8.04_742.4	742.4846	742.4583,742.5464	glycerolipids	DGDG(20:0), MGDG(32:3)	[M+NH4]+	606.81 ± 28.74	224.17 ± 32.49	2.71	<0.05
23	8.04_729.5	729.5348	729.4701, 729.6028	phospholipids	PE(38:8), PS(34:2)	[M+H]+	244.84 ± 19.95	136.24 ± 18.63	1.80	ns
24	8.04_716.4	716.4632	716.4861, 716.4345	phospholipids	LPG(32:5), PI(24:0)	[M+NH4]+	303.01 ± 20.56	30.08 ± 6.45	10.10	<0.0005
25	8.04_575.4	575.4651	575.467	glycerolipids	DG(33:4)	[M+H]+	190.19 ± 23.05	19.17 ± 4.16	9.92	<0.0005
26	8.51_760.5	760.5245	760.5123	phospholipids	PG(34:4)	[M+NH4]+	1675.19 ± 107.08	278.83 ± 55.74	6.01	<0.0005
27	8.51_731.5	731.5506	731.5668, 731.6184	glycerolipids	MGDG(32:0), TG(43:3)	[M+H]+	436.03 ± 22.29	340.56 ± 46.97	1.28	ns
28	8.51_753.5	753.5414	753.5511, 753.6028	glycerolipids	MGDG(34:3), TG(45:6),	[M+H]+	40.47 ± 1.43	42.42 ± 6.82	0.95	ns
29	8.51_744.5	744.5378	744.481, 744.4658,	phospholipids	PG(33:5), PI(26:0)	[M+NH4]+	206.92 ± 21.10	124.11 ± 17.26	1.67	ns
30	8.78_718.4	718.4864	718.4525, 718.5017	phospholipids	PC(30:2(OH), PE(33:2(OH))	[M+NH4]+	423.19 ± 23.43	80.76 ± 16.35	5.24	<0.0005
31	8.78_786.5	786.5374	786.427, 786.5151	glycerolipids	DGDG(24:6), MGDG(36:9)	[M+NH4]+	1252.82 ± 103.59	142.53 ± 25.80	8.79	<0.0005
32	8.78_577.4	577.4322	577.3946, 577.4826	glycerolipids	MGDG(21:0), DG(33:3)	[M+H]+	270.85 ± 30.97	55.24 ± 11.17	4.90	<0.0005
33	8.78_663.4	663.4093	663.4103, 663.4983	glycerolipids	MGDG(28:6), DG(40:9)	[M+H]+	609.72 ± 74.05	507.66 ± 27.32	1.20	ns
34	9.01_766.4	766.4847	766.4583, 766.5464	glycerolipids	DGDG(22:2),MGDG(34:5)	[M+NH4]+	71.82 ± 4.59	31.68 ± 3.10	2.27	<0.05
35	9.01_757.5	757.5558	757.5824, 757.6341	glycerolipids	MGDG(34:1), TG(45:4)	[M+H]+	51.64 ± 2.38	55.87 ± 6.82	0.92	ns
36	9.45_788.5	788.5247	788.5307, 788.6187	glycerolipids	MGDG(36:8), DG(48:11)	[M+NH4]+	201.30 ± 24.97	43.00 ± 8.75	4.68	<0.0005
37	9.45_772.5	772.5918	772.5933, 772.64	glycerolipids	MGDG(34:2), TG(45:5)	[M+NH4]+	99.85 ± 5.59	84.51 ± 9.65	1.18	ns
38	9.45_759.5	759.5811	759.5981, 759.6497	glycerolipids	MGDG(34:0), TG(45:3)	[M+H]+	58.75 ± 1.45	45.27 ± 5.51	1.30	ns
39	9.45_728.5	728.5168	728.5307, 728.5824	glycerolipids	MGDG(31:3), TG(42:6)	[M+NH4]+	31.43 ± 4.19	14.37 ± 1.82	2.19	<0.05
40	9.45_702.4	702.4594	702.4341, 702.4188	phospholipids	PG(30:5), PI(23:0)	[M+NH4]+	84.14 ± 7.79	41.91 ± 2.24	2.00	ns
41	9.68_730.5	730.5211	730.4654	phospholipids	PG(32:5)	[M+NH4]+	243.07 ± 13.67	182.51 ± 19.88	1.33	ns
42	9.68_752.5	752.5024	752.5072, 752.5225	phospholipids	PE(36:6(OH), PC(34:5)	[M+H]+	39.41 ± 3.15	33.47 ± 3.80	1.18	ns
43	9.68_641.4	641.4714	641.5139, 641.4259	glycerolipids	DG(38:6), MGDG(26:3)	[M+H]+	14.66 ± 1.26	7.88 ± 1.39	1.86	ns
44	10.59_566.5	566.5204	566.4779	glycerolipids	DG(31:3)	[M+NH4]+	15.23 ± 1.20	18.96 ± 3.34	0.8	ns
45	10.59_588.4	588.4946	588.4986	glycerolipids	MG(34:6)	[M+NH4]+	1.74 ± 0.39	2.61 ± 0.74	0.67	ns
46	12.75_600.4	600.48	600.5198	glycerolipids	TG(32:0)	[M+NH4]+	17.54 ± 2.35	0.56 ± 0.07	31.5	<0.0005
47	13.63_896.7	896.7049	896.7702	glycerolipids	TG(54:6)	[M+NH4]+	175.31 ± 23.13	12.12 ± 2.46	14.5	<0.0005
48	13.63_870.6	870.6881	870.7545	glycerolipids	TG(52:5)	[M+NH4]+	67.03 ± 5.73	9.21 ± 2.25	7.28	<0.0005
49	13.63_924.7	924.7244	924.8015	glycerolipids	TG(56:6)	[M+NH4]+	8.99 ± 1.70	0.88 ± 0.21	10.2	<0.0005
50	14.03_872.7	872.702	872.7702	glycerolipids	TG(52:4)	[M+NH4]+	729.15 ± 64.44	131.25 ± 17.75	5.56	<0.0005
51	14.03_898.7	898.7174	898.7858	glycerolipids	TG(54:5)	[M+NH4]+	406.46 ± 40.88	51.77 ± 9.89	7.85	<0.0005
52	14.03_877.6	877.6541	877.728	glycerolipids	TG(54:7)	[M+H]+	80.83 ± 4.60	25.55 ± 2.90	3.16	<0.005
53	14.03_846.6	846.6855	846.7545	glycerolipids	TG(50:3)	[M+NH4]+	78.61 ± 4.63	36.13 ± 0.66	2.18	<0.05
54	14.03_900.7	900.7314	900.7076	glycerolipids	TG(55:11)	[M+NH4]+	107.36 ± 12.25	16.55 ± 3.90	6.49	<0.0005
55	14.46_874.7	874.7136	874.6919	glycerolipids	TG(53:10)	[M+NH4]+	2201.20 ± 47.19	1105.18 ± 102.89	2.00	ns
56	14.46_848.7	848.704	848.6763	glycerolipids	TG(51:9)	[M+NH4]+	364.11 ± 10.02	335.58 ± 11.22	3.15	<0.005
57	14.46_880.6	880.6754	880.7389	glycerolipids	TG(53:7)	[M+NH4]+	90.99 ± 4.68	59.83 ± 3.47	1.52	ns
58	14.85_876.7	876.7307	876.7076	glycerolipids	TG(53:9)	[M+NH4]+	1030.29 ± 120.17	1355.54 ± 62.69	0.76	ns
59	14.85_850.7	850.7169	850.6919	glycerolipids	TG(51:8)	[M+NH4]+	66.44 ± 2.03	63.60 ± 3.82	1.04	ns
60	14.85_902.7	902.7433	902.7232	glycerolipids	TG(55:10)	[M+NH4]+	327.72 ± 35.09	207.45 ± 23.05	1.58	ns
61	14.85_881.6	881.699	881.7593	glycerolipids	TG(54:5)	[M+H]+	110.66 ± 3.85	117.84 ± 10.57	0.94	ns
62	14.85_855.6	855.6716	855.7436	glycerolipids	TG(52:4)	[M+H]+	56.90 ± 4.62	62.22 ± 2.34	0.91	ns
63	15.26_878.7	878.7495	878.7232	glycerolipids	TG(53:8)	[M+NH4]+	297.63 ± 42.81	239.60 ± 19.02	1.24	ns
64	15.26_905.7	905.7624	905.7593	glycerolipids	TG(56:7)	[M+H]+	83.15 ± 9.77	53.37 ± 4.27	1.56	ns
65	15.26_883.7	883.7335	883.681	glycerolipids	TG(55:11)	[M+H]+	20.88 ± 1.10	16.90 ± 0.51	1.24	ns
66	15.26_909.6	909.6959	909.7906	glycerolipids	TG(56:5)	[M+H]+	8.24 ± 2.20	5.48 ± 0.14	1.5	ns
67	15.26_852.7	852.7589	852.7076	glycerolipids	TG(51:7)	[M+NH4]+	14.22 ± 1.03	9.74 ± 0.58	1.46	ns

Lipids ID = contains retention time and mass spectrum of the lipid, Input Mass = mass spectra obtained from LC-MS, Matched mass = mass spectra obtained from LIPID MAPS database, Ion = LC-MS run mode, MG = monoglycerol, DG = diglycerol, TG = triglycerol, MGDG = monogalactosyldiacylgylcerol, DGDG = digalactosyldiacylgylcerol, PG = phosphtatidylglycerol, PI = phosphtatidylinositol, PS = phosphtatidylserine, PC = phosphtatidylcholine, PE = phosphtatidylethanolamine, LPG = lysyl-phosphatidylglycerol, LPI = lysophosphatidylinositol, LPA = lysophosphatidic acid. Values with FC (fold change) less than two were considered not significant (ns) by volcano plot statistical analysis.

## Data Availability

The data presented in this study are available in this article.

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
