# Peer review of "Role of Lipids in Phosphine Resistant Stored-Grain Insect Pests Tribolium castaneum and Rhyzopertha dominica"

_insects, 2022, doi:10.3390/insects13090798_

Round 1

Reviewer 1 Report (New Reviewer)

The manuscript reports on the evaluation of the role of lipids in two stored grain pests to resist phosphine. Phosphine resistance is a crucial problem in food security research. In this study, the levels of lipid were compared between phosphine-resistant and susceptible strains of two major coleopteran insect pests Rhyzopertha dominica and Tribolium castaneum. In my opinion, the results of this work will unequivocally contribute to the understanding of the mechanisms of phosphine resistance and provide much additional information for developing strategies for managing phosphine-resistant problems. The manuscript is well organized and well written; however, I have some comments to enhance its quality as follows;

1. Line 15: add “a” before “ higher amount of ……..” 

2. Line 23: “Provide” not “providing” There are some minor English writing mistakes. Please check the whole text.

3. Line 34:You mentioned, “the economic importance of these species derives from contamination of the products….” Is it the economic importance of these two species or damage? It is confusing. It is better to rephrase it.

4. What were the applied photoperiodic settings during the insect rearing? These data need to be inserted.

5. In terms of calculated total lipids, you used chloroform/methanol for extraction and calculated the weight of dry supernatant, but in my opinion, the supernatant still contained non-lipide substances, how to eliminate this interference or did you use other steps for further extraction? Please add more information in this part.

6. Please keep quotations and references consistent in the whole text.

7. How was the LC-MS quantification analysis done? Total Ion or ESI? Peak areas? Since you mentioned fold change between resistant and susceptible species, detection of compounds or method judged detection should be detailed here.

Author Response

Dear Dr. 

In the following, you will find my responses to your valuable comments, which have greatly enhanced the quality of my manuscript. I hope that all comments have been incorporated.

Kind Regards

Author

Reviewer 1

Comments and Suggestions for Authors

The manuscript reports on the evaluation of the role of lipids in two stored grain pests to resist phosphine. Phosphine resistance is a crucial problem in food security research. In this study, the levels of lipid were compared between phosphine-resistant and susceptible strains of two major coleopteran insect pests Rhyzopertha dominica and Tribolium castaneum. In my opinion, the results of this work will unequivocally contribute to the understanding of the mechanisms of phosphine resistance and provide much additional information for developing strategies for managing phosphine-resistant problems. The manuscript is well organized and well written; however, I have some comments to enhance its quality as follows;

  • Author: thank you, your opinion is appreciated.

  1. Line 15: add “a” before “ higher amount of ……..”
  • Author: “a” was added.

  1. Line 23: “Provide” not “providing” There are some minor English writing mistakes. Please check the whole text.
  • Author: 1- the word “providing” was changed to “provide”.

2- The manuscript was further checked and changes made.  

  1. Line 34:You mentioned, “the economic importance of these species derives from contamination of the products….” Is it the economic importance of these two species or damage? It is confusing. It is better to rephrase it.
  • Author: the words “economic importance” was replaced with “damage”.

  1. What were the applied photoperiodic settings during the insect rearing? These data need to be inserted.
  • Author: the information was added to the “insect cultures” section.

  1. In terms of calculated total lipids, you used chloroform/methanol for extraction and calculated the weight of dry supernatant, but in my opinion, the supernatant still contained non-lipide substances, how to eliminate this interference or did you use other steps for further extraction? Please add more information in this part.

Author: 1- good point, thanks for reminding, the information was added.

  1. Please keep quotations and references consistent in the whole text.
  • Author: the manuscript was checked and changes were made.

  1. How was the LC-MS quantification analysis done? Total Ion or ESI? Peak areas? Since you mentioned fold change between resistant and susceptible species, detection of compounds or method judged detection should be detailed here.
  • Author: the information was added, please see “Data processing and analysis” section.

Reviewer 2 Report (New Reviewer)

Phosphine resistance is an important problem facing those wishing to manage insect pests of stored products, so any progress towards understanding how phosphine impacts susceptible or resistant insects has great potential.

The results that resistant insects of two species have more lipids than susceptible strains is very interesting. Considerable research has shown that two major genes (rph1 and rph2) are associated with phosphine resistance. If, as the authors suggest, lipids play a role in resistance, it would be useful to have some discussion at the end of the results and discussion section about how these two research directions might be integrated into a ‘bigger picture’. In other words, are they complementary or mutually exclusive views?

Line 31: ‘…Coleoptera),...’

Lines 39-40: The authors cite only one study in one country to support the statement that phosphine resistance has developed in most insect pests of stored products. This part of the manuscript needs more supporting references.  

Lines 40-41: The authors state that despite ‘numerous studies, the biochemical mechanism of resistance remains unclear, and cite a paper published 22 years ago. The authors should not make statements like this one but instead summarise all relevant published papers.

Lines 80-81: The cited paper is about neonicotinoid resistance not phosphine resistance so seems inappropriate here.

Line 101: ‘One-month-old adults…’

Lines 101-103: It would be useful to have a very brief history of insect strains – where and when they were collected, and whether the strains had undergone any laboratory selection to promote homozygosity for resistance.

Line 108: Why was the wheat frozen for 7 days?

Line 112: ‘…4 days…’

Lines 169-170: ‘…2.5 min…’

Line 305: Give the common name and full scientific name here.

Line 373-375: How will it benefit the development of a management strategy?

Lines 375-376: Why should research be done using isogenic strains?

Author Response

Dear Dr.

In the following, you will find my responses to your valuable comments, which have greatly enhanced the quality of my manuscript. I hope that all comments have been incorporated.

Kind Regards

Author

Reviewer 2

Comments and Suggestions for Authors

Phosphine resistance is an important problem facing those wishing to manage insect pests of stored products, so any progress towards understanding how phosphine impacts susceptible or resistant insects has great potential.

The results that resistant insects of two species have more lipids than susceptible strains is very interesting. Considerable research has shown that two major genes (rph1 and rph2) are associated with phosphine resistance. If, as the authors suggest, lipids play a role in resistance, it would be useful to have some discussion at the end of the results and discussion section about how these two research directions might be integrated into a ‘bigger picture’. In other words, are they complementary or mutually exclusive views?

  • Author: 1- your opinion is appreciated.

2- The information was added to the end of “results and discussion” part, see below:

“Finally, as reported in previous studies, both rph1 and rph2 contribute to phosphine toxicity by damaging fatty acids [9, 13]. In resistant insects, which have homozygous for resistance alleles of rph1 or rph2, the damage of fatty acids is extremely reduced due to the reduction in the sensitivity of cell membranes to reactive oxygen species (ROS) [13]. This, in turn, might lead to an abundance of fatty acids, which are the primary components of lipid formation”.

1- Line 31: ‘…Coleoptera),...’

  • Author: the comma was added.

2- Lines 39-40: The authors cite only one study in one country to support the statement that phosphine resistance has developed in most insect pests of stored products. This part of the manuscript needs more supporting references. 

  • Author: 1- the author added this specific reference as it related to the two species in this study.

2- The word “most” was removed.

3- A new reference was added (reference 7).

3- Lines 40-41: The authors state that despite ‘numerous studies, the biochemical mechanism of resistance remains unclear, and cite a paper published 22 years ago. The authors should not make statements like this one but instead summarise all relevant published papers.

  • Author: good observation; therefore, the statement and the reference were removed from the manuscript.

4- Lines 80-81: The cited paper is about neonicotinoid resistance not phosphine resistance so seems inappropriate here.

  • Author: author believes that this statement doesn’t need to be about phosphine resistance as this is just a general statement to prove that researches are focusing on target sits and metabolic resistance. However, insects are using different strategies to avoid the harmful effect of the pesticides such as using cuticular hydrocarbons to prevent pesticides from entering insect bodies.  

5- Line 101: ‘One-month-old adults…’

  • Author: the word “adult” was added.

6- Lines 101-103: It would be useful to have a very brief history of insect strains – where and when they were collected, and whether the strains had undergone any laboratory selection to promote homozygosity for resistance.

  • Author: the information was added.

7- Line 108: Why was the wheat frozen for 7 days?

  • Author: 1- the wheat was frozen for 7 days to kill any insect eggs or larvae that might be present in the wheat.

2- The phrase “to avoid contamination” was added to manuscript.

8- Line 112: ‘…4 days…’

  • Author: the word “four” was changed to “4”.

9- Lines 169-170: ‘…2.5 min…’

  • Author: the words were changed to “2.5 min”.

10- Line 305: Give the common name and full scientific name here.

  • Author: 1- the common name was added.

2- The scientific name is mentioned for the second time in the manuscript. In the first time was written as a full name Caenorhabditis elegance.

10- Line 373-375: How will it benefit the development of a management strategy?

  • Author:

The results of this research will further insight into the mechanisms of phosphine resistance. In addition to providing valuable information for developing strategies for managing phosphine-resistant problems. Understanding the role of lipids in phosphine resistant stored-grain insects is basic and beneficial to guiding the development strategy for management of phosphine resistance, such as a combination of phosphine fumigation with some technologies or chemicals, which could affect insect lipid metabolism and further lead to synergistic toxicity with phosphine to target insect pests. 

11- Lines 375-376: Why should research be done using isogenic strains?

  • Author:

1- Using two species of stored grain insects gave us more reliable results; in both species, resistance strains exhibited higher levels of lipids.

2- Considering these results as a starting point for future studies, the author recommended conducting further studies on phosphine susceptible and resistant strains that come from the same origin (one male and one female) to avoid the confusion that the differences in lipids might be due to genetic background.

This manuscript is a resubmission of an earlier submission. The following is a list of the peer review reports and author responses from that submission.

Round 1

Reviewer 1 Report

This work attempts to test insect resistance to phosphine. It presents a case on the role of lipids and sugars as antagonists to phosphine.  In general, the whole paper requires extensive check for English grammar, spellcheck and logical flow of ideas. The research experimental design is also not clear.

Overall recommendation is to reject paper, carry out major revisions for the paper before submission to any journal for review. The experimental design needs to be stated clearly in terms of the hypothesis, the number of replicates and treatments, what variables were tested and statistical analyses conducted.

I have following comments for the authors to consider as they rewrite.

Title: Rephrase the title; suggestion “Lipids and sugars antagonize phosphine toxicity in stored grain insects”

Abstract: Rewrite the abstract for ease of reading. It is hard for the reader to follow what this study was about. Suggestion: First sentence- short introduction; second sentence- objective/justification; third sentence- Methods; fourth to sixth sentence: key findings; final sentence: conclusion/take home message

Keywords: This section should not list the same terms as the title i.e., phosphine, stored product insects, lipids and sugars. It should have 3-5 words that appear repeatedly in the article but are not in the title.

Introduction: The introduction should be rewritten with each paragraph covering a specific topic. The whole introduction talks mostly of lipids (line 56 to 71 are too superfluous about lipids, the information in these two paragraphs can be summarized in paragraph 2 and the space used to discuss other topics instead). There is very little information on sugars, the two stored product pests studied and previous research on the mechanism of phosphine resistance in these 2 pests. At the end of the introduction add a clear objective and well-articulated hypothesis for the study.

Methods: Bring line 93 at the beginning of section 2.1 for ease of flow of information and also helps list the activities in the order that they were done. Also cite the references for the methods used for this paper

Results: State the results and statistical significance of the results in the text. Simply showing the figures is not enough.

Author Response

Dear Dr.

I would like to thank you for your highly perceptive comments, which significantly guided me to improve the quality of my manuscript. In the following are my response and point by point addressed viewers comments. I have also attached a copy of revised manuscript. I hope that I have incorporated all comments and advice.

Kind Regards

Reviewer: This work attempts to test insect resistance to phosphine. It presents a case on the role of lipids and sugars as antagonists to phosphine. In general, the whole paper requires extensive check for English grammar, spellcheck and logical flow of ideas. The research experimental design is also not clear.

Authors:

  • A native English speaker has checked the English.

  • The manuscript flow was adjusted to reflect the reviewers comments.

Reviewer: Overall recommendation is to reject paper, carry out major revisions for the paper before submission to any journal for review. The experimental design needs to be stated clearly in terms of the hypothesis, the number of replicates and treatments, what variables were tested and statistical analyses conducted.

Authors:

  • The experimental design: all the experiments were done according to the "completely randomized design (C.R.D.)".

b- The research hypothesis was stated at the end of the introduction lines 85-97:

  • We updated the aim of the study; please see below:

"This study aimed to evaluate the role of lipids and sugar in stored grain insects tolerate to phosphine".

  • Number of replicates and treatments: as explained in the methodology, two species were tested in separate experiments to determine the lipid's content. Each species has two strains. The number of the replicates was stated in the "Data processing and analysis" section. In addition, both the number of treatments and replicates is on the figures and from figure titles.
  • Variables were tested and statistical analyses conducted: authors explained in "Data processing and analysis" the statistical analysis, all statistical analysis was carried out using MetaboAnalyst 4.0, please refer to the link https://www.metaboanalyst.ca/MetaboAnalyst/upload/StatUploadView.xhtml, and the lines 226-234.
  • Also, authors stated the data analysis in each title of the figures; please refer to the figure titles.
  • We have added more information regarding figures 1, 2, 3, 6 and 7.

Reviewer: Title: Rephrase the title; suggestion "Lipids and sugars antagonize phosphine toxicity in stored grain insects"

Authors:

  • Thanks for the suggestion; the author believes this is not a suitable title for the current manuscript as resistant insects not only use lipids but are also using different strategies to avoid phosphine damage.

Reviewer: Abstract: Rewrite the abstract for ease of reading. It is hard for the reader to follow what this study was about. Suggestion: First sentence- short introduction; second sentence- objective/justification; third sentence- Methods; fourth to sixth sentence: key findings; final sentence: conclusion/take home message

Authors:

  • thank you for the suggestion; abstract was rewritten to meet the reviewer suggestion:

Introduction: Insects rely on lipids as an energy source needed to perform various bioactivities. Flight, growth, diapause, and metamorphosis are essential processes that cannot be performed without a sufficient amount of lipids.

Objective: This study evaluated the role of lipids and sugar in stored grain insects to tolerate the toxicity of phosphine

Methodology: Phosphine resistant and susceptible strains of the two main stored-grain insects, T. castaneum and R. dominica, were analysed using liquid chromatography-mass spectroscopy (LC-MS) to determine their lipid contents and sugar.

Key finding: The lipids and sugar, which significantly contribute to survival, were evaluated, and the ef-fect of short and long-term starving treatment on insect survival was compared. Rapid depletion of sugar was detected from resistant strains both T. castaneum and R. dominica after one day of starving. In contrast, the lipids content was relatively high for both species until seven days of starving. The resistant insects of both species have a higher amount of lipids than the susceptible insects. Significant variance ratios between the resistant and susceptible strains of T. castaneum were observed for most of the lipids ranging with glycerolipids from 1.13 to 53.10 fold and phospholipids from 1.05 to 20.00 fold. In comparison, the fold changes for R. dominica between the resistant and susceptible strains were 1.04 to 31.50 fold and from 1.04 to 10.10 fold for glycerolipids and phospholipids respectiverly.

Conclusion: Lipids offer a consistent source of energy, in addition to providing a suitable environment to protect the mitochondria from phosphine. Hence, it was proposed through this study that the lipid content of phosphine resistant and susceptible strains of T. castaneum and R. dominica could play an important role in tolerance of phosphine

Reviewer: Keywords: This section should not list the same terms as the title i.e., phosphine, stored product insects, lipids and sugars. It should have 3-5 words that appear repeatedly in the article but are not in the title.

Authors:

  • Thank you for the suggestion; we have made changes by removing some words and focusing most used words. See below:

Keywords: "phosphine; insect resistance; T. castaneum; R. dominica; glycerolipids; phospholipids".

Reviewer: Introduction: The introduction should be rewritten with each paragraph covering a specific topic. The whole introduction talks mostly of lipids (line 56 to 71 are too superfluous about lipids, the information in these two paragraphs can be summarized in paragraph 2 and the space used to discuss other topics instead). There is very little information on sugars, the two stored product pests studied and previous research on the mechanism of phosphine resistance in these 2 pests. At the end of the introduction add a clear objective and well-articulated hypothesis for the study.

Authors:

  • We have added more information about insect species: lines 32-37
  • Information about sugar was added: lines 55-62.
  • The two paragraphs were summarised combined in one section.
  • Information about the mechanism of phosphine resistance from previous literature is in lines 41-54. No specific mechanism regarding the two species used in this study.

Reviewer: Methods: Bring line 93 at the beginning of section 2.1 for ease of flow of information and also helps list the activities in the order that they were done. Also cite the references for the methods used for this paper.

Authors:

  • Line 93 was brought to the beginning of section 2.1.
  • Citations were added (references 10, 38, 39 and 40)

8- Reviewer: Results: State the results and statistical significance of the results in the text. Simply showing the figures is not enough.

Authors:

  • The statistical significances were added; please see below:

lines 296-310.

Reviewer 2 Report

The manuscript by Alnajim et al describes differences in lipid and sugar content between phosphine susceptible and resistant strains of red flour beetle and lesser grain borer as analyzed by LC/MS.  It has long been suggested that metabolic differences could drive phosphine resistance in some stored product species.  The authors of the current manuscript attempt to argue that differences in lipid and sugar composition are important for resistance to phosphine; however, the experimental design is not suitable to support these claims.  Several examples of where this study falls short are listed below along with line by line edits:

  1. The authors show that sugars are quickly depleted in resistant red flour beetles and lesser grain borers after starvation, but that lipid levels deplete more slowly. The authors claim this is probably important for resistance, but do not show similar data for susceptible individuals.

  1. The authors claim that the differences in lipid and sugar levels between the susceptible and resistant populations drive resistance. However, the comparison is purely descriptive and the differences in lipid and sugar levels could be ancillary to the physiological processes that are underlying resistance in these insects. 

  1. A better design could have involved exposing the susceptible and resistant individuals to phosphine and performing the lipid/sugar analysis on individuals that survived the fumigation. These changes could also be compared to control insects.

  1. In addition, it is likely that not all resistant individuals survived the fumigation. The authors could have tested individuals from the resistant population that survived the fumigation for lipid sugar content and compared them to individuals from the resistant population that did not survive to see if differences in lipid/sugar content were associated with survival. 

  1. It is unclear if the authors have genotyped these populations to determine whether they carry any known resistance alleles for phosphine (DLD, for example). It is also unclear if any of these known mutations have been previously associated with changes in lipid or sugar levels previously.  The authors should present more background in their introduction about what is known regarding the physiological differences between resistant and susceptible information.  Some information is presented, but I think they need to go deeper.  I have provided some specific suggestions.

  1. There is some information in both the introduction and discussion that is unclear and needs more details for support. Specific examples are listed below.

Lines 13-14:  I don’t think it is entirely correct to state that the study assessed the role of lipids in phosphine resistance in these two insect species.  While there is a correlation between differences in lipid content between the resistant and susceptible strains, the authors have not yet demonstrated causation.

Lines 16-17:  I also don’t think it’s entirely correct to state that the effects of lipid composition and sugar content on survival was evaluated because the authors simply compared lipids and sugars between the susceptible and resistant strains.

Line 19:  In contrast, the lipid content remained relatively high in the resistant strain (?) until….

Lines 23-24:  What are these fold changes in relation to?  Resistant:susceptible?  This needs to be clearly stated in the abstract.

Line 26:  It’s hard to know based on this study if these changes are really essential factors in resistance.  These differences could be due to the differences in physiology between susceptible and resistant strains, but they may be directly or indirectly related to the mechanism of resistance.

Line 34:  I would rephrase this:  Phosphine is among the most successful fumigants that is currently approved to control….

Line 36:  Despite numerous studies,

Line 41:  this phrasing does not make much sense:  are an essential aspect of comprehending the resistance phenomenon.  Suggest replacing with “are strongly associated with the resistance phenomenon to this fumigant.”

Lines 42-43:  Some more information about the proposed mechanism of DLD-mediated resistance should be provided in the introduction.  In addition, there are other mutations that interact with DLD that mediate the strength of this resistance.  Those should also be discussed and would provide necessary context to support the authors’ statements on lines 43-46.  How do these mutations impact physiology of organisms that carry them?

Line 47:  I am not entirely sure what the authors mean when they say that the composition of lipids occurs naturally.  Lipid biosynthesis is an organic process that can be affected by numerous biotic and abiotic factors.  This is common knowledge, but perhaps that authors need to be more specific with these statements to show how it relates to the correct topic.

Line 50:  this phrase does not make much sense, please revise:  components in continuing the insects on this planet.  I think the authors mean that the ability to store fat is essential for insects to adapt to their environment and undergo normal development and reproduction….

Line 51:  lipids are considered the main energy reserves, but they also have other functions….

Line 56:  Lipids that can be used for energy are stored in the form of triglycerides.

Line 74:  More information should be provided about the proposed mechanisms behind the different forms of phosphine resistance and how they differ from one another (or overlap). 

Line 79:  instead of using the word “assume,” you should rephrase this statement to indicate that you are testing the hypothesis that lipid and sugar composition differ between susceptible and resistant populations.

Line 87:  Were either of these populations tested genetically for mutations that have been previously associated with phosphine resistance?  Is that how the authors knew that these populations were susceptible and resistant to phosphine or were those classifications based on earlier research?

Line 87:  Is this how the cultures were initiated in the lab?  Approximately 3000 adult insects of approximately 2-3 days in age (post-eclosion?)

Line 88: which consisted of wheat flour/yeast 12:1 ratio for….  All cultures were maintained in 2L jars….

Line 90:  Adults were allowed to mate and lay eggs for a period of four days, after which they were removed and the remaining….

Line 92:  to keep insects with similar age ranges together.

Lines 93-97:  Was wheat used to rear R. dominca and flour used for T. castaneum?  Please specify.

Line 108:  while the LC50 for the resistant strains was…

Lines 112-119: In addition to the solvents and supplies used to prepare the extracts, the authors should also describe the procedure that was used to create the extracts.  Or this this information presented in section 2.5?  If so, please move section 2.5 before the section on lipid and sugar analysis (2.4).

Line 124:  Separation of lipid compounds was performed…

Line 136:  Separation of sugar compounds was achieved on….

Line 174:  replace patch with batch

Line 178:  The resulting data matrix consisted of

Line 182:  was restricted to

Line 187:  Was there any confidence threshold applied to remove spurious database identifications from the analysis (ie, only spectrum matches with confidence values above XX% were considered for identification). 

Line 214: I am unclear how the lipid and sugar content in response to starvation can be linked to the survival ability of the insects in each strain.  The experiment is set up as a descriptive experiment.  If the authors wanted to provide more conclusive evidence linking changes in lipids and sugars to ability to survive phosphine fumigation, perhaps individuals from both populations that either survived the fumigation at the LC50 dose or died after exposure could be compared to see if there are any differences in lipid/sugar contents among the insects that survived the treatment (if possible). 

Line 216:  Sugar levels quickly reduced in the resistant strains after starvation.  What about the susceptible strains?

Line 229:  Again, with the design for this study, it is really not possible to determine whether these differences are predictably involved in phosphine resistance.

Line 230:  If significant variance was observed, what do the chromatograms in figures 4 and 5 represent?  Were those derived from a single pool of 15 individuals from each strain? 

Figure 6:  There are no axis labels for X on the graph depicting the glycerolipids.  Are the retention times for those the same as are depicted for the phospholipids?  Also, were the differences tested statistically and should asterisks be added to indicate significant differences?

Lines 253-265:  Any specific glycerolipids or phospholipids that were significantly higher in the resistant strains?  What about compounds that were higher in resistant individuals of both species?  I would have liked to see more of a comparison here to see if there were any similarities. 

Line 273-274:  This should be stated as a more of a hypothesis since this was not directly tested.

Line 285:  It is unclear how reference 6 supports this statement.  Trehalose was not measured in that study at all.  In addition, it is unclear if this statement applies to all situations or just to the general physiological conditions that were studied in the single study that was referenced by the authors.

Line 293:  Again, there is nothing in the experimental design that indicates that the lipids protect against harmful effects of phosphine.  The authors just showed that lipid were elevated in the resistant insects, which just be associated with the changes in metabolism that are needed for phosphine resistance.

Line 294:  This is because lipids can provide a constant source of energy needed to overcome various stressors.

Lines 295-297:  Incomplete sentence.  In addition, it’s unclear how lipids protect against toxicity to mitochondria.  Carbon from lipids still needs to be converted to ATP by mitochondrial enzymes or through anaerobic mechanisms.

Lines 295-302:  This information could be reorganized and consolidated to make a better point.  I think the authors are trying to state that insects with high energy demands are more susceptible to phosphine because of the increased mitochondrial activity levels that are necessary to sustain energy production. It is unclear the way this paragraph is written how lipids could help overcome this situation as the carbon from these compounds may still have to be converted to ATP by the mitochondria.

Lines 303-306:  Again, by what process are lipids converted to ATP?

Line 308:  which, along with glycogen, is one of the main sources of stored energy

Line 314:  some information about how proline can be used for energy by insects (via oxidation) should be included here.

Lines 343-344:  This statement needs a citation.

Author Response

Dear DR. 

I would like to thank you for your highly perceptive comments, which significantly guided me to improve the quality of my manuscript. In the following are my response and point by point addressed viewers comments. I have also attached a copy of revised manuscript. I hope that I have incorporated all comments and advice.

Kind Regards

Reviewer 2

The manuscript by Alnajim et al describes differences in lipid and sugar content between phosphine susceptible and resistant strains of red flour beetle and lesser grain borer as analyzed by LC/MS. It has long been suggested that metabolic differences could drive phosphine resistance in some stored product species. The authors of the current manuscript attempt to argue that differences in lipid and sugar composition are important for resistance to phosphine; however, the experimental design is not suitable to support these claims. Several examples of where this study falls short are listed below along with line by line edits:

Reviewer: The authors show that sugars are quickly depleted in resistant red flour beetles and lesser grain borers after starvation, but that lipid levels deplete more slowly. The authors claim this is probably important for resistance, but do not show similar data for susceptible individuals.

Authors:

  • The manuscript was adjusted to reflect the reviewers comments.

Reviewer: The authors claim that the differences in lipid and sugar levels between the susceptible and resistant populations drive resistance. However, the comparison is purely descriptive and the differences in lipid and sugar levels could be ancillary to the physiological processes that are underlying resistance in these insects.

 Authors:

  • Authors didn't claim that sugar or lipids drive resistance. However, resistant insects are using different mechanisms to avoid the harmful effect of phosphine. Example, literature acknowledge that the respiration rate of resistant insects is usually lower than the respiration rate of susceptible insects, but nobody is saying that low respiration leads to resistance. This strategy makes resistant insects to absorb less amount of phosphine compared to susceptible insects. Similarly, authors believe that resistant insects using lipids to face up the lack of energy when they are treated with phosphine.

Reviewer: A better design could have involved exposing the susceptible and resistant individuals to phosphine and performing the lipid/sugar analysis on individuals that survived the fumigation. These changes could also be compared to control insects.

 Authors:

  • Good observation, there are several reasons to not using phosphine:
  • The parents of the resistant strains used in this study were probably exposed to or survived phosphine many time many times in the field before its resistance rise to this level.
  • After transferring to the insect culture room, insects were reared in "food safety and biosecurity laboratories" for a long period. During this period, resistant populations were purified using phosphine to preserve strains with high resistance factors.
  • At the time of conducting the study, phosphine wasn't used as the aim of the study was to know what the metabolic differences between resistant and susceptible are without any pressure caused by treating with phosphine.
  • The comparison in every experiment should be conducted under same condition of using same concentrations of phosphine. However, resistant and susceptible are very different in their response to phosphine. Using very low concentration of phosphine (LC50 of susceptible) might not affect the resistant insects (will not kill any individuals of the resistant population). In comparison, using moderate concentration (LC50 of resistant) might cause death for individuals of susceptible insects. According to our results which was showed in the manuscript, 0.009 mg/L is required to kill 50% of the susceptible strain of dominica, in comparison, to kill 50% of the resistant insects we need 1.042 mg/L. That means if we use 0.009 mg/L to kill 50% of the susceptible insects, the resistant will not affect at all.
  • Using different phosphine concentrations with the resistant and susceptible insects will make the comparison unfair, as we will put the strains under different level of pressure.

Reviewer: In addition, it is likely that not all resistant individuals survived the fumigation. The authors could have tested individuals from the resistant population that survived the fumigation for lipid sugar content and compared them to individuals from the resistant population that did not survive to see if differences in lipid/sugar content were associated with survival.

 Authors:

  • At this stage of our study, we only aimed to determine the metabolic differences between resistant and susceptible are without any pressure caused by treating with phosphine. The authors are planning to conduct more experiments regarding the effect of using phosphine.

Reviewer: It is unclear if the authors have genotyped these populations to determine whether they carry any known resistance alleles for phosphine (DLD, for example). It is also unclear if any of these known mutations have been previously associated with changes in lipid or sugar levels previously. The authors should present more background in their introduction about what is known regarding the physiological differences between resistant and susceptible information. Some information is presented, but I think they need to go deeper. I have provided some specific suggestions.

Authors:

  • Thanks for the suggestion, we carried out bioassays to determine the resistance level of the strains used in this study. This method was accredited by FAO organization (reference 38) as a standard method to determine the resistance level of some stored grain insect species, including castaneum and R. dominica.
  • The authors added literature related to physiological differences between susceptible and resistant strains; as examples, please see references 5, 8, 9, 10, 35, 36, 37, 39, 42, 43, 44, 45, and 56.

Reviewer: There is some information in both the introduction and discussion that is unclear and needs more details for support. Specific examples are listed below.

Authors:

  • All suggestions are taken into consideration.

Point by point review

Reviewer: Lines 13-14:  I don't think it is entirely correct to state that the study assessed the role of lipids in phosphine resistance in these two insect species. While there is a correlation between differences in lipid content between the resistant and susceptible strains, the authors have not yet demonstrated causation.

Authors:

  • There is a difference between lipid content between RS and SS and hence based on our data lipid might also be playing a role in the tolerance of Ph3 in resistance strain”.  

Reviewer: Lines 16-17:  I also don't think it's entirely correct to state that the effects of lipid composition and sugar content on survival was evaluated because the authors simply compared lipids and sugars between the susceptible and resistant strains.

Authors:

  • The starving study was only conducted to confirm the importance of lipids compared to sugar in resistant strains (as they adapted to resist lack of the energy) if they are exposed to severe conditions like starving for a long period.
  • Author did compared lipids between the susceptible and resistant strains.

Reviewer: Line 19:  In contrast, the lipid content remained relatively high in the resistant strain (?) until….

Authors:

  • The sentence was completed, please see line 19.

Reviewer: Lines 23-24:  What are these fold changes in relation to? Resistant:susceptible? This needs to be clearly stated in the abstract.

Authors:

  • The information was updated, please see lines 22-26

Reviewer: Line 26: It's hard to know based on this study if these changes are really essential factors in resistance. These differences could be due to the differences in physiology between susceptible and resistant strains, but they may be directly or indirectly related to the mechanism of resistance.

Authors:

  • As previously mentioned, resistant insects use different strategies to resist phosphine. Based on our data, we believe that the existence of a high amount of lipids provides resistant insects with an advantage to face the energy deficiency caused by phosphine.

Reviewer: Line 34:  I would rephrase this:  Phosphine is among the most successful fumigants that is currently approved to control….

Authors:

  • Thank you, the sentence was rephrased according to the suggestion, line 38.

Reviewer: Line 36:  Despite numerous studies,

Authors:

  • The sentence was changed.

Reviewer: Line 41:  this phrasing does not make much sense:  are an essential aspect of comprehending the resistance phenomenon. Suggest replacing with "are strongly associated with the resistance phenomenon to this fumigant."

Authors:

  • Thanks for the suggestion, the sentence was rewritten, lines 45-47.

  • Reviewer: Lines 42-43: Some more information about the proposed mechanism of DLD-mediated resistance should be provided in the introduction. In addition, there are other mutations that interact with DLD that mediate the strength of this resistance. Those should also be discussed and would provide necessary context to support the authors' statements on lines 43-46. How do these mutations impact physiology of organisms that carry them?

Authors:

  • The information was added, lines 49-52.

Reviewer: Line 47:  I am not entirely sure what the authors mean when they say that the composition of lipids occurs naturally. Lipid biosynthesis is an organic process that can be affected by numerous biotic and abiotic factors. This is common knowledge, but perhaps that authors need to be more specific with these statements to show how it relates to the correct topic.

Authors:

  • The sentence was removed from the text.

Reviewer: Line 50:  this phrase does not make much sense, please revise:  components in continuing the insects on this planet. I think the authors mean that the ability to store fat is essential for insects to adapt to their environment and undergo normal development and reproduction….

Authors:

  • The sentence was changed, lines 64-65.

Reviewer: Line 51:  lipids are considered the main energy reserves, but they also have other functions….

Authors:

  • More information were added, see lines 65-66.

Reviewer: Line 56:  Lipids that can be used for energy are stored in the form of triglycerides.

Authors:

  • The sentence was changed according to the "reviewer 3" recommendation, please see lines 70-72. "Lipids are stored in the form of triglycerides (TGs) inside the fat bodies responsible for meeting the energy requirements of insects by storing large quantities of triglycerides [24]".

Reviewer: Line 74:  More information should be provided about the proposed mechanisms behind the different forms of phosphine resistance and how they differ from one another (or overlap).

Authors:

  • Information was added, lines 86-92.

Reviewer: Line 79:  instead of using the word "assume," you should rephrase this statement to indicate that you are testing the hypothesis that lipid and sugar composition differ between susceptible and resistant populations.

Authors:

  • The sentence was changed, please see lines 93-96.

Reviewer: Line 87:  Were either of these populations tested genetically for mutations that have been previously associated with phosphine resistance? Is that how the authors knew that these populations were susceptible and resistant to phosphine or were those classifications based on earlier research?

Authors:

  • We used standard FAO accredited method (reference 38) to determine the resistance level of stored grain insect species, including castaneum and R. dominica.

Reviewer: Line 87:  Is this how the cultures were initiated in the lab? Approximately 3000 adult insects of approximately 2-3 days in age (post-eclosion?)

Authors:

  • The sentence was changed.

Reviewer: Line 88: which consisted of wheat flour/yeast 12:1 ratio for….  All cultures were maintained in 2L jars….

Authors:

  • Sentences were changed according to the reviewer's recommendation.

Reviewer: Line 90:  Adults were allowed to mate and lay eggs for a period of four days, after which they were removed and the remaining….

Authors:

  • The sentence was changed.

Reviewer: Line 92:  to keep insects with similar age ranges together.

Authors:

  • The sentence was changed.

Reviewer: Lines 93-97:  Was wheat used to rear R. dominca and flour used for T. castaneum? Please specify.

Authors:

  • The sentence was changed.

Reviewer: Line 108:  while the LC50 for the resistant strains was…

Authors:

  • The sentence was changed.

Reviewer: Lines 112-119: In addition to the solvents and supplies used to prepare the extracts, the authors should also describe the procedure that was used to create the extracts. Or this this information presented in section 2.5? If so, please move section 2.5 before the section on lipid and sugar analysis (2.4).

Authors:

  • Section 2.5 was moved.

Reviewer: Line 124:  Separation of lipid compounds was performed…

Authors:

  • The sentence was changed according to the reviewer's recommendation.

Reviewer: Line 136:  Separation of sugar compounds was achieved on….

Authors:

  • The sentence was changed according to the reviewer's recommendation.

Reviewer: Line 174:  replace patch with batch

Authors:

  • The word "patch" was replaced with "batch".

Reviewer: Line 178:  The resulting data matrix consisted of

Authors:

  • The sentence was changed.

Reviewer: Line 182:  was restricted to

Authors:

  • The sentence was changed according to the reviewer's recommendation.

Reviewer: Line 187:  Was there any confidence threshold applied to remove spurious database identifications from the analysis (ie, only spectrum matches with confidence values above XX% were considered for identification).

Authors:

  • No confidence threshold was applied, however, only the closest spectra were selected for compound identification.

Reviewer: Line 214: I am unclear how the lipid and sugar content in response to starvation can be linked to the survival ability of the insects in each strain. The experiment is set up as a descriptive experiment. If the authors wanted to provide more conclusive evidence linking changes in lipids and sugars to ability to survive phosphine fumigation, perhaps individuals from both populations that either survived the fumigation at the LC50 dose or died after exposure could be compared to see if there are any differences in lipid/sugar contents among the insects that survived the treatment (if possible).

Authors:

  • Resistant and susceptible are very different in their response to phosphine. Low concentration of phosphine might not affect the resistant insects (will not kill any individuals of the resistant population). In comparison, using moderate or high concentrations might cause death for all individuals of susceptible insects. According to our results as shown in the manuscript, 0.009 mg/L is required to kill 50% of the susceptible strain of dominica, in comparison to 1.042 mg/L , to kill 50% of the resistant insects... So doses required would be different for susceptible strain and resistant strains to achieve LC 50, and this will add another variable factor to the experiment. Hence, we avoided taking insects post fumigation.

Reviewer: Line 216:  Sugar levels quickly reduced in the resistant strains after starvation. What about the susceptible strains?

Authors:

  • Authors only conducted the sugar study to confirm factor (sugar or lipids) that the insects will use to survive when they are under strong pressure.

Reviewer: Line 229:  Again, with the design for this study, it is really not possible to determine whether these differences are predictably involved in phosphine resistance.

Authors:

  • According to our data, we believe that lipids have role in surviving phosphine effect even if the difference was as consequence of phosphine resistance.

Similar to our idea, previous studies confirmed that as consequence of phosphine resistance, the respiration rate of resistant insects is lower than the susceptible insects. The lower respiration rate benefits the resistant insects by absorbing less amount of phosphine through respiration.  

Reviewer: Line 230:  If significant variance was observed, what do the chromatograms in figures 4 and 5 represent? Were those derived from a single pool of 15 individuals from each strain?

Authors:

  • The chromatograms in figures 4 and 5 show the differences of the total lipids using the LC-MS technology. They are derived from a single pool of 15 individuals from each strain separately, but the analysis was repeated with four different biological sets.

Reviewer: Figure 6:  There are no axis labels for X on the graph depicting the glycerolipids. Are the retention times for those the same as are depicted for the phospholipids? Also, were the differences tested statistically and should asterisks be added to indicate significant differences?

Authors:

  • Both glycerolipids and phospholipids have the same labels for in the graphs to reduce the number the graphs in the manuscript.
  • Asterisks were added to indicate significant differences.

Reviewer: Lines 253-265:  Any specific glycerolipids or phospholipids that were significantly higher in the resistant strains? What about compounds that were higher in resistant individuals of both species? I would have liked to see more of a comparison here to see if there were any similarities.

Authors:

  • The glycerolipids or phospholipids that were significantly higher in resistant individuals of both species were added, please see from lines 305-311.

  • Reviewer: Line 273-274: This should be stated as a more of a hypothesis since this was not directly tested.

Authors:

  • The sentence was rephrased. Lines 320-322

Reviewer: Line 285:  It is unclear how reference 6 supports this statement. Trehalose was not measured in that study at all. In addition, it is unclear if this statement applies to all situations or just to the general physiological conditions that were studied in the single study that was referenced by the authors.

Authors:

  • The reference was corrected to the right one.
  • Authors believe this applies to all situations.

Reviewer: Line 293:  Again, there is nothing in the experimental design that indicates that the lipids protect against harmful effects of phosphine. The authors just showed that lipid were elevated in the resistant insects, which just be associated with the changes in metabolism that are needed for phosphine resistance.

Authors:

  • The authors believe that phospholipids may reduce the penetration of phosphine to the cells, causing more exclusion of phosphine, as phospholipids contribute to the mitochondrial membrane environment. They have a substantial role in the mitochondrial respiratory chain by affecting the physical properties of the mitochondrial membrane.

Reviewer: Line 294:  This is because lipids can provide a constant source of energy needed to overcome various stressors.

Authors:

  • The sentence was rephrased according to the reviewer's recommendation.

Reviewer: Lines 295-297:  Incomplete sentence. In addition, it's unclear how lipids protect against toxicity to mitochondria. Carbon from lipids still needs to be converted to ATP by mitochondrial enzymes or through anaerobic mechanisms.

Authors:

  • The sentence was rewritten only to indicate Phospholipids.
  • Phospholipids are essential components of cellular and semi cellular membranes. Therefore, they may reduce the penetration of phosphine to the cells, causing more exclusion of phosphine.
  • Phospholipids also contribute to the mitochondrial membrane lipid environment. They have a substantial role in the mitochondrial respiratory chain by affecting the physical properties of the mitochondrial membrane.

Reviewer: Lines 295-302:  This information could be reorganized and consolidated to make a better point. I think the authors are trying to state that insects with high energy demands are more susceptible to phosphine because of the increased mitochondrial activity levels that are necessary to sustain energy production. It is unclear the way this paragraph is written how lipids could help overcome this situation as the carbon from these compounds may still have to be converted to ATP by the mitochondria.

Authors:

  • The sentence was rephrased in lines 340-342.

Reviewer: Lines 303-306:  Again, by what process are lipids converted to ATP?

Authors:

  • By this statement, the authors wanted to say that lipids can help resistant insects recover when exposed to phosphine. As explained before, resistant insects are using different strategies to survive phosphine; when resistant insects are treated with phosphine, about a 1-2 weeks recovery period is needed for the insects to recover their bioactivities to the normal level. During this period, mitochondria stay active but not as normal. The existence of a high level of lipids may help insects to recover fast.

Reviewer: Line 308:  which, along with glycogen, is one of the main sources of stored energy

Authors:

  • The sentence was rephrased.

Reviewer: Line 314:  some information about how proline can be used for energy by insects (via oxidation) should be included here.

Authors:

  • Information was added. Lines 360-361

Reviewer: Lines 343-344:  This statement needs a citation.

Authors:

  • The sentence has been cited.

Reviewer 3 Report

It is an interesting study. Please revise some sections based on the comments provided.

Author Response

Dear DR. 

I would like to thank you for your highly perceptive comments, which significantly guided me to improve the quality of my manuscript. In the following are my response and point by point addressed viewers comments. I have also attached a copy of revised manuscript. I hope that I have incorporated all comments and advice.

Kind Regards

Reviewer 3

1- Reviewer: Line 56 Lipids are stowed in the form of triglycerides. Should be stored.

Authors:

  • The words was changed to "stored".

2- Reviewer: Line 56-57 should be "Lipids are stored in the form of triglycerides (TGs) inside the fat bodies responsible for meeting the energy requirements of insects by storing large 57 quantities of triglycerides [13]".

Authors:

  • The sentence was rephrased according to the reviewer's recommendation. “Lipids are stored in the form of triglycerides (TGs) inside the fat bodies responsible for meeting the energy requirements of insects by storing large quantities of triglycerides [26]”.

3- Reviewer: Line 80-81: Why have you selected Tribolium 80 castaneum and Rhyzopertha dominica species of all the stored product pests. You need to stress the importance of these two species to justify your selection.

Authors:

  • to show the importance of the two species, we have added the following to the introduction: lines 32-37.

4- Reviewer: Lines 87-88: Narrow age of insects (2-3 days), What do you mean by narrow age??? Rewrite.

Authors:

  • the sentence was rewritten.

5- Reviewer: Line 98: prepatration of phosphine gas for determination of resistance factor

should be :Prepetration of phosphine gas for determination of resistance factor.

Authors:

  • The word "prepetration" was changed to "preparation".

6- Reviewer: Line 124: LIPIDS why it is capitalized.

Authors:

  • the word "LIPIDS" was rewritten using lowercase "Lipids"

7- Reviewer: Lines 208-209: The resistant insects of both T. castaneum and R. dominica showed to 208 have more lipids than the susceptible insects.

Should be

The resistant strains of both T. castaneum and R. dominica species demonstrated a significantly greater amount of lipids compared to the susceptible strains of both species (Fig. 1).

Author:

  • Thank you, the sentence was rewritten according to the reviewer recommendation.

8- Reviewer: According to Figure 1, The resistant strains of both T. castaneum and R. dominica species demonstrated a significantly greater amount of lipids compared to the susceptible strains of both species; My question is why is the amount of lipids lower in R. dominica compared with T. castaneum? Of course, there is a difference in the amount of lipids in the resistant and the susceptible strains of each species. Does the low amount of lipids in R. dominica effects its growth and development as well as defense from phosphine compared with T.castaneum with high levels of lipids.

Authors:

  • Good question, no comparison was conducted between dominica and T. castaneum as this wasn't the aim of the study; however, the main reason, in my opinion, adult R. dominica is smaller than T. castaneum, . The total bodyweight of 15 adult insects of R. dominica is about 0.01881±0.00134 g. in contrast, 15 adult insects of T. castaneum weighted about 0.02459±0.00117 g.

Round 2

Reviewer 3 Report

The manuscript looks fine except for minor changes as following:

Lines  14, 32-33- write full names of species: Tribolium castaneum and Rhyzopertha dominica.

2.2   Preparation

Lines 417-418 rewrite the sentence:

This had their importance as the main sources of energy and their contribution to the cell walls.

Author Response

Dear reviewer 

Please see below my response to the comments. 

Thank you

Author 

Reviewer 

The manuscript looks fine except for minor changes as following:

Authors: thank you.

Reviewer: Lines  14, 32-33- write full names of species: Tribolium castaneum and Rhyzopertha dominica.

Authors: the species were written in full name.

Reviewer: 2.2   Preparation

Authors: the word was rewritten in uppercase.

3- Reviewer: Lines 417-418 rewrite the sentence:

This had their importance as the main sources of energy and their contribution to the cell walls.

Authors: the sentence was rewritten. Please see 421-422.